# Population Fitness of *Eupeodes corollae* Fabricius (Diptera: Syrphidae) Feeding on Different Species of Aphids

**DOI:** 10.3390/insects13060494

**Published:** 2022-05-24

**Authors:** Shanshan Jiang, Hui Li, Limei He, Kongming Wu

**Affiliations:** 1State Key Laboratory for Biology of Plant Diseases and Insect Pests, Institute of Plant Protection, Chinese Academy of Agricultural Sciences, Beijing 100193, China; jiangss97@163.com (S.J.); lihuilh521@163.com (H.L.); 2Institute of Urban Agriculture, Chinese Academy of Agricultural Sciences, Chengdu 610200, China; helimei91@163.com

**Keywords:** *Eupeodes corollae*, fecundity, life history, flight ability, aphid species

## Abstract

**Simple Summary:**

As a candidate for controlling aphid populations, *Eupeodes corollae* Fabricius also performs a pollination function which is critical to agricultural systems. We evaluated the population life tables and flight performance of *E. corollae* fed on three prey species (*Aphis craccivora* Koch, *Myzus persicae* Sulzer and *Megoura japonica* Matsumura) to select suitable aphid species for keeping *E. corollae* indoors. The results showed that *E. corollae* completed development and reproduction on these three aphid species while achieving the shortest generation time, the maximum fecundity, and the highest intrinsic natural growth rate and flight ability on *M. japonica*. Our study indicated that *M. japonica* is the most suitable prey for *E. corollae*, providing a basis for utilizing the ecological service function of the hoverfly.

**Abstract:**

*Eupeodes corollae* Fabricius, as one of the most common beneficial predatory insects in agricultural ecosystems, provides pollination and biological control services that help improve crop yield and maintain biodiversity. However, systematic research is needed on the species of aphids used for propagation. To develop highly fit populations of the important insect predator and crop pollinator, *E. corollae*, for research and commercial use, further research is needed to develop the most nutritious diet and efficient propagation methods. Here, the fitness of *E. corollae* was assessed in the laboratory after larvae were fed an aphid diet of *Aphis craccivora* Koch, *Myzus persicae* Sulzer or *Megoura japonica* Matsumura. The larval survival rate on *M. japonica* was significantly lower than on *A. craccivora* and *M. persicae*. The developmental duration for larvae (7.6 d) and pupae (6.9 d) was longest on *A. craccivora*. The pupal emergence rate on *A. craccivora* (98.0%) was significantly higher than on the other two, and lowest (64.7%) on *M. japonica*. On *A. craccivora*, *M. persicae*, and *M. japonica*, respectively, the generation time was 24.85 d, 23.12 d and 21.05 d; the value for the intrinsic rate of natural increase was 0.19, 0.20, and 0.21; and the value for the finite rate of increase was 1.21, 1.22, and 1.23. For flight variables, *E. corollae* attained the fastest velocity and longest distance and duration on *M. japonica*. The *M. japonica* diet, thus, provided the shortest generation time, the highest intrinsic rate of natural increase and finite rate of increase, the maximum fecundity and the greatest flight ability. Thus, to improve the survival rate of *E. corollae* larvae, *A. craccivora* or *M. persicae* can be used to feed newly hatched larvae, and *M. japonica* can be used for second- and third-instar larvae. These results provide a theoretical basis for feeding *E. corollae* and optimizing its ecosystem services.

## 1. Introduction

Syrphidae, also known as flower flies or hoverflies, comprise almost 6000 species in 200 genera [1,2] and are found all over the world except for some remote islands in Antarctica and the Pacific Ocean (such as Hawaii) [3,4]. In China, 465 species in 80 genera of Syrphidae are known [5]. Hoverfly larvae may be saprophages, carnivores or other types of feeders [6,7,8]. About one third of hoverfly species have carnivorous larvae that prey on whiteflies, thrips and aphids. Thus, the larvae of these species are natural enemies that might be used to control aphids in agroecosystems [9]. The third-instar larvae of *E. corollae* consume about 60 cabbage aphids per day per individual and 300 cabbage aphids in total [10]. In addition, most hoverfly adults visit flowers and are the second most important pollinators after bees. They are even more effective than bees at pollinating individual plants such as *Paphiopedilum dianthum* [11,12,13]. Their pollination efficacy, however, varies depending on the crop species, season and location [14]. For example, hoverfly pollination increased fennel yield by 104.9% [15] and parsley seed yield nearly 3-fold compared with a control group not pollinated by hoverflies [16].

*Eupeodes corollae* Fabricius (Diptera: Syrphidae), a common hoverfly species worldwide, is a promising candidate for aphid control and pollination [17]. The larvae are carnivorous and feed on a variety of aphid species and small lepidopteran larvae [18,19]. In one study, larvae consumed 64.7 wheat aphid individuals in a field, providing a control rate of 74.8% [20]. When 8-day-old larvae of *E. corollae* were released at a benefit-to-harm ratio of 1:10, the number of aphids decreased by 85.9% after 8 days [21]. As pollinators, *E. corollae* adults can significantly increase crop yields. The fruit yield and seed set of sweet peppers pollinated by *E. corollae* increased, respectively, by 390% and 395% compared to the control group not pollinated by hoverflies [22].

For the study and development of hoverflies for agricultural use, however, a large supply of high-quality insects at different stages is needed. The few studies on large scale propagation have focused on aspects of adult nutrition, prey species and host plants. The larvae of *Episyrphus balteatus* (De Geer) and *Eupeodes bucculatus* (Rondani) that fed on an artificial diet mainly composed of drone honeybee brood powder (DP) grew and developed normally, but did not lay eggs as adults [23,24]. At present, hoverflies are all reared using aphids as food; however, this process has many problems such as self-mutilation by larvae, obligate diapause of adults, and mating difficulty [25]. In addition, there is a lack of research on various species of aphids and their effects on fitness variables of *E. corollae*.

Hoverflies are excellent candidates to control aphid populations due to their high predation and reproductive capacity [26,27], but the development of solid or liquid diets has been difficult, because *E. corollae* larvae do not eat dead aphids [28]. Here, we used the major aphid pests *Aphis craccivora* Koch, *Myzus persicae* Sulzer and *Megoura japonica* Matsumura (Homoptera: Aphididae) [29], which readily reproduce in greenhouses [21], as diets for *E. corollae* in the laboratory. We then assessed larval developmental, reproductive and flight variables for *E. corollae* to select the best nutritional source as a theoretical basis for rearing *E. corollae* indoors. This work will advance research on syrphids.

## 2. Materials and Methods

### 2.1. Aphid Collection and Culture

Aphids (*A. craccivora*, *M. persicae* and *M. japonica*) were collected from the experimental field at the Langfang Experimental Station of Agricultural Sciences in Hebei Province, China (CAAS; 39°30′29″ N, 116°36′8″ E) in 2016. *Aphis craccivora* and *M. japonica* were reared on bean plantlets, and *M. persicae* were reared on pea plantlets until more than 30 generations in the greenhouse at 25 ± 1 °C, 50 ± 5% RH and 16:8 (L:D) h. All plantlets were grown in nutrient soil and vermiculite in plastic boxes (50 × 40 × 18 cm).

### 2.2. Hoverfly Collection and Culture

In June 2018, 25 adults of *E. corollae* (♀:♂ = 15:10) were also collected from the field at the Langfang Experimental Station of Agricultural Sciences (CAAS; 39°30′29″ N, 116°36′8″ E). Adults were fed with a mixture of pollen (rape: corn = 3:1) and 10% *v/v* honey water in nylon gauze cages (30 × 40 × 50 cm). Broad bean plantlets infested with mixed aphids (*A. craccivora*:*M. persicae*:*M. japonica* ≈ 1:1:1) were placed in the cages for laying eggs. Larvae were fed on *A. craccivora* on bean plantlets until 10 consecutive generations in laboratory at 25 ± 1 °C, 50 ± 5% RH and 16:8 (L:D) h.

### 2.3. Life Table Study for E. corollae

Newly hatched larvae of *E. corollae* were raised with *A. craccivora*, *M. persicae* or *M. japonica*, individually, in petri dishes (3.5 cm diameter × 1 cm height), with 100 aphids added daily until pupation. On the third day after pupation, each pupa was weighed on an electronic scale (Sartorius CPA225D; Suzhou Sainz Instrument Co., Ltd., Suzhou, China). The development (whether they molted) and survival of *E. corollae* larvae were observed and recorded every day with a stereo microscope (XTL-165-VT, Phenix Optical Co., Ltd., Shangrao, China) until they pupated or died. Each diet treatment consisted of 30 eggs or newly hatched larvae and was replicated three times. Larval survival rate, pupation rate, and emergence rate were calculated as Larval survival rate = No. of mature larvae/No. of tested larvae × 100; Pupation rate = No. of pupae/No. of mature larvae × 100; and Emergence rate = No. of adult/No. of pupae × 100.

Emerging adults (♀:♂ = 1:1) were reared in 20 × 30 × 45 cm cages (200-mesh nylon) with broad bean plantlets infested with mixed aphids (*A. craccivora*:*M. persicae*:*M. japonica* ≈ 1:1:1), and also with a mixture of pollen (rape:corn = 3:1), and 10% honey water provided daily. *Eupeodes corollae* eggs on broad bean plantlets were counted and transferred to a petri dish (9 cm diameter × 1 cm height) to observe hatching. The plantlets were replaced every day with fresh ones until *E. corollae* adults died.

The data on developmental and reproductive variables, generation time (*T*), finite rate of increase (*λ*), intrinsic rate of natural increase (*r*), and net reproductive rate (*R*_0_) of the experimental populations of *E. corollae* were calculated according to the following formulas [30,31]:$$l_{x}=\sum_{j=1}^{m}S_{xj}$$
$$m_{x}=\frac{\sum_{j=1}^{m}S_{xj}f_{xj}}{\sum_{j=1}^{m}S_{xj}}$$
$$\sum_{x=0}^{\infty }l_{x}m_{x}e^{-r\left(x+1\right)}=1$$
$$R_{0}=\sum_{x=0}^{\infty }l_{x}m_{x}$$
$$T=\frac{\ln R_{0}}{r}$$
$$\lambda =e^{r}$$
where *x* is the time interval in days (d); *l_x_* is the survival rate of *E. corollae* from egg to *x* days old; *m* is the number of stages; *S_xj_* is the survival rate of *E. corollae* from egg development to *x* days old and developmental stage *j*; *f_xj_* is the age-specific fecundity at age *x*; and *m_x_* is the average population fecundity from egg to *x* days old.

### 2.4. Flight Tests for E. corollae

Flight variables (duration, distance, velocity) of 5-day-old adults were measured using a flight mill (FXMD-24-USB, Jiaduo Science Industry and Trade Co., Ltd., Hebi, China) as described previously [32,33]. The mesothorax was attached to the arm of a flight mill using 502 glue (Shenzhen Jinsan second Adhesive Co., Ltd., Shenzhen, China). The mill was kept in a climate chamber (MGC-450HP, Shanghai Yiheng Scientific Instrument Co., Ltd., Shanghai, China) at 25 ± 1 °C with 50 ± 5% RH. Flights started at approximately 20:00 and ran for 10 h in the dark. More than 30 individuals were successfully tested for each prey treatment.

### 2.5. Data Analysis

Differences in the flight ability variables and pupal mass were analyzed using non-parametric Kruskal–Wallis test followed by a Bonferroni-adjusted significance test for pairwise comparison at the 0.05 significance level. Differences in larval survival rate, pupation rate, emergence rate and hatching rate of eggs were determined using a one-way analysis of variance (ANOVA) followed by Tukey’s honestly significant difference (HSD), with proportional data first arcsine square-root-transformed to meet the assumptions of normality and heteroscedasticity. Differences in the duration of developmental stages between genders were determined using a Student’s *t*-test, and differences in survival curves of *E. corollae* after feeding on different aphids were analyzed using a log rank test. A paired bootstrap test with 100,000 replications was used for the precise estimation of the mean and standard error among the demographic parameters and developmental duration of the 3 diet treatments. All tests were performed in the program SPSS version 25 (IBM, Armonk, NY, USA), except life table variables, which were calculated and differentially analyzed in TWO-SEX-MSChart (Chi 2019) and plotted in OriginPro 2021 (OriginLab Corporation, Northampton, MA, USA).

## 3. Results

### 3.1. Development of E. corollae Fed on Different Aphids

All the *E. corollae* larvae that fed on the three aphid species completed development, but the larval and pupal stages varied, and development differed between sexes in all prey treatments (Table 1). More specifically, there were significant differences in the duration of the first instar, second instar, third instar and larval stage (first–third instar) between the different aphid treatments. The duration of larvae (7.6 d) and pupae (6.9 d) after feeding on *A. craccivora* was significantly longer than for those fed on *M. persicae* or *M. japonica*. The first instar (1.4 d) was the shortest on *M. persicae*. The second instar (1.8 d), third instar (3.0 d) and pupal (6.1 d) stages were the shortest on *M. japonica*. The duration of the combined larva–adult stage and the adult lifespan of *E. corollae* did not differ significantly among on the different aphid diets. On *A. craccivora,* the duration of first-instar larvae (*t* = 2.816, *df* = 62, *p* = 0.007), larval stage (first–third instar) (*t* = 2.398, *df* = 62, *p* = 0.020), pupae (*t* = 2.667, *df* = 62, *p* = 0.012), adult (*t* = 4.172, *df* = 62, *p* < 0.001) and larva–adult (*t* = 4.728, *df* = 62, *p* < 0.001) of females were significantly longer than those of males. On *M. persicae*, the duration of first instar (*t* = 2.703, *df* = 39, *p* = 0.014), pupae (*t* = 6.658, *df* = 39, *p* < 0.001), adult (*t* = 3.162, *df* = 39, *p* = 0.003) and larva–adult (*t* = 3.792, *df* = 39, *p* = 0.001) of females were significantly longer than those of males. On *M. japonica*, the first instar (*t* = −3.135, *df* = 25, *p* = 0.005) of females was significantly shorter than that of males.

The aphid species greatly affected the larval survival rate (*F*_2,6_ = 12.022, *p* = 0.008), pupal emergence (*F*_2,6_ = 14.853, *p* = 0.005), and pupal mass (Kruskal–Wallis, *H* = 26.607, *p* < 0.001) of *E. corollae* (Table 2). The larval survival rate of *E. corollae* on *M. persicae* (84.4%) was significantly higher than that on *M. japonica* (54.4%), but the pupal emergence rate was significantly higher on *A. craccivora* (98.0%) than on *M. persicae* (73.9%) and *M. japonica* (64.7%).

The aphid species also significantly affected the generation time (*T*) of *E. corollae*. On *A. craccivora*, *T* was the longest, and the finite rate of increase (*λ*) and intrinsic rate of natural increase (*r*) were the lowest. On *M. japonica*, however, *T* was the shortest, and *λ* and *r* were the highest. The order of population growth from highest to lowest on the three aphid diets was *M. japonica* > *M. persicae* > *A. craccivora* (Table 3).

The survival rate of *E. corollae* larvae, pupae and adults was the lowest on *M. japonica* (0.52 for egg development to second-instar larvae, 0.51 for third-instar larvae, 0.41 for pupae and 0.15 for adults) (Figure 1). The survival rate was highest for egg development to second-instar larvae (0.88) and third-instar larvae (0.84) on the *M. persicae* diet, but highest on *A. craccivora* for egg development to pupae (0.72) and adult (0.35).

The *l_x_* curves for *E. corollae* also differed significantly among the three aphid diets (*x*^2^ = 26.594, *df* = 2, *p* < 0.001). For the *M. japonica* diet, the *l_x_* curve of *E. corollae* decreased by 0.7 from day 4 to day 16, which was a faster decrease than for *A. craccivora* and *M. persicae*. From day 20 to day 30 (adult stage), the *l_x_* curve of *E. corollae* that fed on *A. craccivora* decreased rapidly, indicating a higher mortality rate for adults. The peak spawning day for females was day 23 on *A. craccivora*, day 21 on *M. persicae* and day 18 on *M. japonica*, as shown on the *l_x_m_x_* curves (Figure 2).

### 3.2. Reproduction of E. corollae Fed on Different Aphids

The aphid diets also greatly affected pre-oviposition, total pre-oviposition and number of ovipositions of *E. corollae* (Table 4). On *A. craccivora*, the total pre-oviposition was the longest (20.3 d) and the oviposition number was the least (305.9 eggs per female). On the contrary, the total pre-oviposition was the shortest (16.6 d) and the oviposition number was greatest (596.5 eggs per female) on *M. japonica*.

### 3.3. Flight Performance of E. corollae Fed on Different Aphids

There were no significant differences in flight variables (flight duration, flight distance, flight velocity). Adults flew the fastest (0.25 m/s), and the longest distance (624.0 m) and duration (2521.9 s) when they fed on *M. japonica*. The maximum cumulative flight distance (1622.7 m), maximum speed (0.38 m/s), and maximum cumulative duration (6217.2 s) were achieved on *A. craccivora* (Table 5).

## 4. Discussion

*Eupeodes corollae* depresses aphid populations in ecosystems [34,35,36]. Our present results show that *E. corollae* completed development and reproduced after consuming the three aphid species. *M. japonica* consumption, however, yielded the highest fecundity, finite rate of increase, intrinsic rate of natural increase and flight performance, and the shortest generation time. Therefore, among the three test species, *M. japonica* is considered to be the most suitable for laboratory propagation of *E. corollae*.

The predatory activity of larvae is closely related to the type, quantity and quality of prey; higher fitness afforded by the prey can be assessed by higher survival rate and faster development of the larval stage of the predator [37,38,39]. The larval stage of *E. corollae* was shortest when the larvae fed on *M. japonica*, but the larval survival rate was significantly reduced, as also found by Xiong and Dong [21]. Newly hatched *E. corollae* larvae are small and inactive and, thus, feed only on smaller prey. The body of *M. japonica* is larger than that of *A. craccivora* and *M. persicae*; thus, *M. japonica* might have been able to escape predation, contributing to the lower survival rate of *E. corollae* larvae on *M. japonica*. The curves for the age–stage survival rate of various instar larvae after all prey treatments overlapped because individual development was inconsistent, coincident with overlapping generations in the field [40].

Pupal performance and adult fecundity mainly depend on the nutrients consumed by the larvae [41,42,43]. *Eupeodes corollae* attained the shortest pupal duration and total pre-oviposition and the most ovipositions on *M. japonica*, as well as the lowest pupal emergence rate, as also reported in a previous study [21]. On *A. craccivora*, *E. corollae* attained the longest pupal duration, greatest total pre-oviposition and the fewest ovipositions, as well as the highest pupal emergence rate and pupal mass, which may be associated with aphid nutrient content. High-quality prey improves the performance (e.g., pupal mass, fecundity, longevity) of hoverflies [44], as also found for predatory ladybirds [45,46].

Based on the life table variables (such as *R*_0_, *r* and *λ*) for the *E. corollae* population to evaluate the suitability of a prey species [47], *r* was greater than 0, and *λ* was greater than 1 for all three aphid diets, revealing that they are suitable prey for *E. corollae*. Further, the highest *r* and *λ* (albeit not significantly different) for the *E. corollae* population among the three aphid diets was obtained on *M. japonica*, which suggested that *M. japonica* was the most suitable among the three aphids. The searching and predation activities of *E. corollae* larvae improve with increasing larval age [48]. Because the survival rate of *E. corollae* larvae on *M. japonica* is significantly lower than on *A. craccivora* or *M. persicae*, we recommend that *A. craccivora* or *M. persicae* be used to feed newly hatched larvae and that *M. japonica* be used for second- and third-instar larvae.

For practical applications, the flying ability of predatory insects affects their colonization and reproduction which, in turn, affects their control range and efficacy [49]. Hoverfly activities such as foraging, finding mates and spawning sites are dependent on their flight abilities. As the environment changes, hoverflies—especially the predatory hoverflies, which have poorer flight ability compared with other lepidopteran insects [50,51]—migrate long distances to reproduce [52], requiring good nutrition for the energy needed. Our results showed that the flying performance of *E. corollae* adults was significantly affected by the species of aphid consumed during the larval stage, which helps with *E. corollae* forecasting in the field.

Our systematic evaluation of the development, reproduction and flight ability of *E. corollae* laboratory populations that fed on different species of aphids determined the best aphid species to rear different stages of *E. corollae*. These findings will serve as the basis of large-scale rearing of hoverflies with high fitness and efficacy as biocontrol agents and pollinators. The field environment is much more complex and harsher than the laboratory, and the occurrence of hoverfly populations is highly coincident with that of other natural enemies such as ladybugs and lacewings [53]. Therefore, the fitness of the laboratory *E. corollae* populations reared on aphid diets now need to be evaluated in the field.

## 5. Conclusions

Herein, our research proved that *E. corollae* larvae can complete development and reproduction on *A. craccivora*, *M. persicae* and *M. japonica*. There is no significant difference in the larval–adult development period of *E. corollae* on the three aphid species. *Eupeodes corollae* attained the highest fecundity, intrinsic rate of natural increase and flight ability and the shortest generation time on *M. japonica*. In conclusion, *M. japonica* is considered to be the most suitable prey for the indoor rearing of *E. corollae* among the three tested aphids; *Aphis craccivora* or *M. persicae* can be used to feed newly hatched larvae of *E. corollae*, and *M. japonica* can be used for second- and third-instar larvae.

## Figures and Tables

**Figure 1 insects-13-00494-f001:**
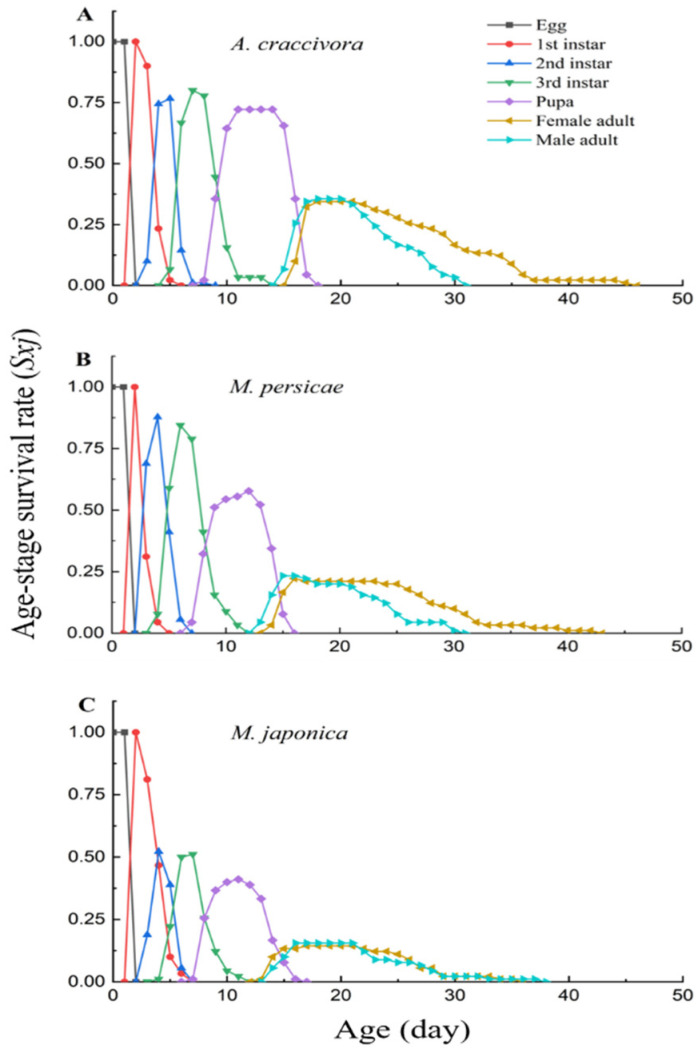
Survival rate curves for *E. corollae* after larvae fed on *A. craccivora* (**A**), *M. persicae* (**B**) and *M. japonica* (**C**) in the laboratory.

**Figure 2 insects-13-00494-f002:**
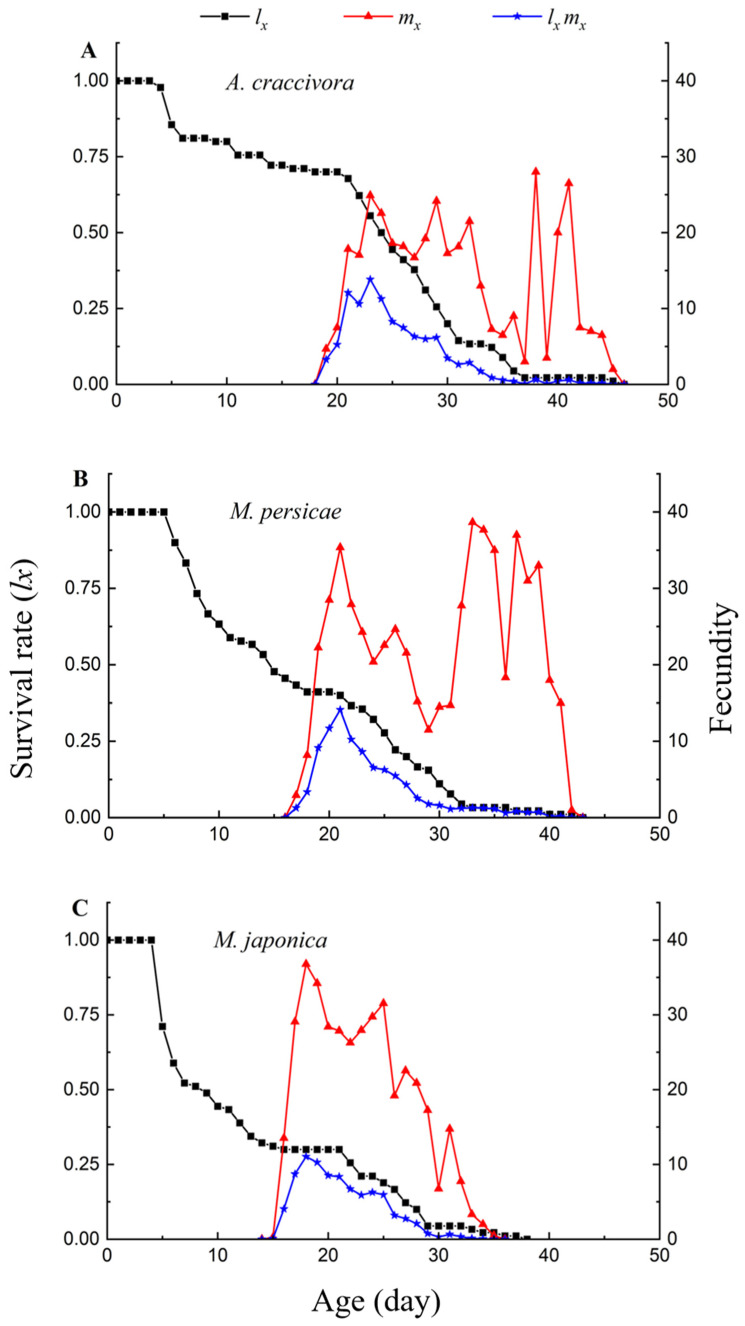
Population age-specific fecundity (*m_x_*), population age-specific survival rate (*l_x_*) and population age-specific maternity (*l_x_m_x_*) for *E. corollae* after larvae fed on *A. craccivora* (**A**), *M. persicae* (**B**) and *M. japonica* (**C**) in the laboratory.

**Table 1 insects-13-00494-t001:** Stage durations for *E. corollae* after larvae fed on different aphid species.

Stage	Sex	Stage Duration (d)					
		*A. craccivora*	*n*	*M. persicae*	*n*	*M. japonica*	*n*
Egg	♀ + ♂	2.0 ± 0.00 a	90	2.0 ± 0.00 a	90	2.0 ± 0.00 a	90
	♀	2.0 ± 0.00 A	31	2.0 ± 0.00 A	20	2.0 ± 0.00 A	13
	♂	2.0 ± 0.00 A	33	2.0 ± 0.00 A	21	2.0 ± 0.00 A	14
1st instar	♀ + ♂	2.0 ± 0.06 a	75	1.4 ± 0.06 b	90	1.8 ± 0.09 a	55
	♀	2.1 ± 0.06 A	31	1.5 ± 0.18 A	20	1.5 ± 0.18 B	13
	♂	1.8 ± 0.09 B	33	1.0 ± 0.00 B	21	2.2 ± 0.11 A	14
2nd instar	♀ + ♂	2.1 ± 0.03 a	72	1.9 ± 0.06 b	76	1.8 ± 0.07 b	49
	♀	2.1 ± 0.04 A	31	1.7 ± 0.14 A	20	1.8 ± 0.15 A	13
	♂	2.1 ± 0.04 A	33	1.8 ± 0.09 A	21	1.6 ± 0.13 A	14
3rd instar	♀ + ♂	3.5 ± 0.07 a	65	3.6 ± 0.13 a	55	3.0 ± 0.13 b	41
	♀	3.6 ± 0.09 A	31	3.1 ± 0.08 A	20	2.8 ± 0.10 A	13
	♂	3.5 ± 0.10 A	33	3.4 ± 0.13 A	21	3.0 ± 0.23 A	14
Larval stage (1st–3rd instar)	♀ + ♂	7.6 ± 0.09 a	65	6.7 ± 0.16 b	55	6.7 ± 0.16 b	41
	♀	7.8 ± 0.10 A	31	6.4 ± 0.11 A	20	6.2 ± 0.20 A	13
	♂	7.4 ± 0.14 B	33	6.1 ± 0.16 A	21	6.8 ± 0.23 A	14
Pupae	♀ + ♂	6.9 ± 0.04 a	64	6.3 ± 0.07 b	41	6.1 ± 0.07 b	27
	♀	7.0 ± 0.00 A	31	6.7 ± 0.10 A	20	6.1 ± 0.10 A	13
	♂	6.8 ± 0.07 B	33	6.0 ± 0.00 B	21	6.1 ± 0.10 A	14
Adult	♀ + ♂	11.8 ± 0.69 a	64	12.5 ± 0.84 a	41	12.4 ± 0.77 a	27
	♀	14.5 ± 1.11 A	31	14.9 ± 1.25 A	20	13.3 ± 0.91 A	13
	♂	9.3 ± 0.55 B	33	10.1 ± 0.89 B	21	11.6 ± 1.21 A	14
Egg–adult	♀ + ♂	28.3 ± 0.70 a	64	27.1 ± 0.88 a	41	27.1 ± 0.83 a	27
	♀	31.3 ± 1.07 A	31	30.0 ± 1.26 A	20	27.7 ± 1.08 A	13
	♂	25.5 ± 0.59 B	33	24.3 ± 0.86 B	21	26.6 ± 1.26 A	14

Values are mean ± SE. Different lowercase letters on the same row mean significant differences (paired bootstrap test, *p* < 0.05). Means for male and female at the same developmental stage in same column followed by different capital letters differed significantly (Student’s *t*-test, *p* < 0.05).

**Table 2 insects-13-00494-t002:** Variables and pupal mass of *E. corollae* after larvae fed on different aphid species.

Variable	*A. craccivora*	*n*	*M. persicae*	*n*	*M. japonica*	*n*
Larval survival rate (%)	80.0 ± 5.77 a	3	84.4 ± 2.94 a	3	54.4 ± 4.84 b	3
Pupation rate (%)	89.9 ± 5.52 a	3	72.8 ± 6.78 a	3	83.1 ± 7.62 a	3
Emergence rate (%)	98.0 ± 1.96 a	3	73.9 ± 6.29 b	3	64.7 ± 4.05 b	3
Hatching rate of egg (%)	90.0 ± 5.09 a	3	88.89 ± 4.01 a	3	82.2 ± 6.76 a	3
Mass (g)						
♀	0.0564 ± 0.0133 A	31	0.0317 ± 0.0009 A	20	0.0315 ± 0.0006 A	13
♂	0.0332 ± 0.0005 A	33	0.0307 ± 0.0009 A	21	0.0309 ± 0.0009 A	14
♀ + ♂	0.0444 ± 0.0066 a	64	0.0312 ± 0.0006 b	41	0.0312 ± 0.0005 b	27

Values are mean ± SE. Different lowercase letters on the same row reflect significant differences (one-way ANOVA, Tukey’s HSD, *p* < 0.05). Different capital letters in the same column indicate significant differences (Student’s *t*-test, *p* < 0.05) in pupal mass between male and female fed on the same aphid.

**Table 3 insects-13-00494-t003:** Life table variables for *E. corollae* after larvae fed on different aphid species.

Prey	*A. craccivora*	*M. persicae*	*M. japonica*
*T*	24.85 ± 0.34 a	23.12 ± 0.41 b	21.05 ± 0.33 c
*λ*	1.2052 ± 0.0089 a	1.2167 ± 0.0137 a	1.2335 ± 0.0182 a
*r*	0.1866 ± 0.0074 a	0.1961 ± 0.0113 a	0.2098 ± 0.0149 a
*R* _0_	105.38 ± 19.65 a	96.39 ± 24.90 a	86.16 ± 24.42 a

Values are mean ± SE. Different lowercase letters on the same row reflect significant differences (paired bootstrap test, *p* < 0.05).

**Table 4 insects-13-00494-t004:** Fecundity variables for *E. corollae* after larvae fed on different aphid species.

Prey	*A. craccivora*	*n*	*M. persicae*	*n*	*M. japonica*	*n*
No. of ovipositions	305.9 ± 36.23 b	31	433.7 ± 73.80 ab	20	596.5 ± 74.85 a	13
Oviposition duration (d)	10.2 ± 1.11 a	30	11.6 ± 1.18 a	19	11.0 ± 0.89 a	13
Total pre- oviposition (d)	20.3 ± 0.18 a	30	18.9 ± 0.29 b	19	16.6 ± 0.37 c	13
Pre-oviposition (d)	3.5 ± 0.11 a	30	3.8 ± 0.26 a	19	2.2 ± 0.17 b	13

Values are mean ± SE. Different lowercase letters on the same row reflect significant differences (paired bootstrap test, *p* < 0.05).

**Table 5 insects-13-00494-t005:** Flight variables for *E. corollae* after larva fed on different aphids.

Prey		*A. craccivora* (*n* = 32)	*M. persicae* (*n* = 36)	*M. japonica* (*n* = 37)
Duration (s)	Max	6217.2	4305.6	5079.6
	Mean ± SE	2442.3 ± 240.3 a	2037.4 ± 151.3 a	2521.9 ± 186.8 a
Distance (m)	Max	1622.7	950.2	1407.3
	Mean ± SE	582.4 ± 69.2 a	448.2 ± 35.4 a	624.0 ± 52.0 a
Velocity (m/s)	Max	0.38	0.33	0.36
	Mean ± SE	0.23 ± 0.01 a	0.22 ± 0.01 a	0.25 ± 0.01 a

Values are mean ± SE. Different lowercase letters on the same row reflect significant differences (Kruskal–Wallis test, *p* < 0.05).

## Data Availability

All data analyzed in this study are included in this article.

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
