# Peer review of "Population Fitness of Eupeodes corollae Fabricius (Diptera: Syrphidae) Feeding on Different Species of Aphids"

_insects, 2022, doi:10.3390/insects13060494_

Round 1
Reviewer 1 Report
Dear editor,
In their manuscript entitled “Population fitness of Eupeodes corollae Fabricius (Diptera: Syrphidae) feeding on different species of aphids” [insects-1731448-peer-review-v1] Jiang and colleagues evaluated the development, reproduction and flight ability of E. corollae laboratory populations that fed on different species of aphids (A. craccivora, M. persicae and M. japonica). The study was motivated by the fact that E. corolla is one of the most common beneficial predatory insects in agricultural ecosystems providing pollination and biological control services, but importantly systematic research on the species of aphids used for propagation is largely missing. Experiments from this study showed that was no significant difference in the larval-adult development period of E. corollae on the three aphid species. Eupeodes corollae attained the highest fecundity, intrinsic rate of natural increase, flight ability and the shortest generation time on M. japonica. Although the pupal emergence rate was on lowest on M. japonica, authors suggest M. japonica as the most suitable prey for indoor rearing of E. corollae among the tested aphids. These results lay the basis of large-scale rearing of hoverflies with high fitness and efficacy as biocontrol agents and pollinators.
The manuscript is well written, the research design is appropriate and the results support most of the conclusions. Overall the findings are of high interest to the field, although there are some issues to clarify before I can finally recommend publication in Diversity. Please see below for details.
- line 68: add “as adults” behind “but did not lay eggs [23, 24]” since authors refer to the larval stage only in this sentence
- line 77: at the beginning for is mentioned twice, delete accordingly
- line 93: write “b” in broad in as capital
- line 124: correct “fllight” to “flight”
- lines 151-153: a verb is missing in this sentence
- line 388: insert space after “armyworm”
- line 396 and 399 : insert space after “Rondani”
Author Response
Reviewer 1:
- line 68: add “as adults” behind “but did not lay eggs [23, 24]” since authors refer to the larval stage only in this sentence;
Response: Based on your comment, we have added “as adults” behind “but did not lay eggs [23, 24]”. Thanks. (Line 71)
- line 77: at the beginning for is mentioned twice, delete accordingly;
Response: Revised as suggested and we have deleted “for”. Thanks. (Line 80)
- line 93: write “b” in broad in as capital;
Response: Revised as suggested and we have changed “broad” to “Broad”. Thanks. (Line 96)
- line 124: correct “fllight” to “flight”;
Response: Based on your comment and we have corrected “fllight” to “flight”. Thanks. (Line 132)
- lines 151-153: a verb is missing in this sentence;
Response: Revised as suggested and we have changed to “More specifically, there were significant differences in the duration of the 1st instar, 2nd instar, 3rd instar and larval stage (1st–3rd instar) between the different aphid treatments”. (Lines 160-163)
- line 388: insert space after “armyworm”;
Response: Thanks for your suggestion. we have inserted space after “armyworm”. (Line 399)
- line 396 and 399 : insert space after “Rondani”
Response: Thanks for your suggestion. we have inserted space after “Rondani”. (Lines 407-410)

Reviewer 2 Report
The research investigates life variables of E. corollae on three aphids diets.
This is well-written paper with appropriate experimental design for the stature of Insect Journal.
I have few suggestion of this work that you could read below.
Abstract:
L24 : second time that use "systematic" maybe try another one
L32-33 : do not use more than 2 significant numbers after "."
M&M :
L83-87 : How many generations before experimentation were reared the aphids from the fields collection ?
L90-94 : Same question for the hoverfly species ? How many individuals did you collect during these trials or in other word how many individuals constituted your starting population ?
L98 : were raised individually in Petri dishes? please specify it.
L100 : how many aphids ?
L138-139 : Prior to use the ANOVA, do you test the normality of each variables ? If not, you might violate assumption conditions of the ANOVA and in this case needs to pass to a non parametric test (such as Kruskal-Wallis).
L139 : Justify why to use the "Tukey’s honestly significant difference (HSD)" test ?
L140 : On which variables do you apply proportional data, is it on mx ?
Results :
L153 : "different aphid treatments"
L158 : "not"
L160 : "pupae" ?
Table 1 : it seems confusing with the different letters after mean± SE, could you please better explain than you did in the end of this table ?
Table 2 : I did not understand to which groups or individuals the number n = 3 is referring to ? Could you specify this ?
Figure 1 : Can we have SEM on each point of all the survival curves ?
Figure 2 : Can we have SEM on each point of all the survival curves ?
Table 3-4-5 : To be coherent with the first tables, it would be interesting to invert lines and columns of these three tables.
Discussion :
L312-315 : I completely agree with this statement. In the conclusion part and the abstract, this could be put forward more.
L325-329 : This part could move in the intro part, it does not add some improvements at this part of the discussion, it could also be deleted from the discussion.
Author Response
Reviewer 2:
Abstract:
L24 : second time that use "systematic" maybe try another one
Response: Revised as suggested and we have changed “systematic” to “further”. (Line 24)
L32-33 : do not use more than 2 significant numbers after "."
Response: As suggested, life table parameters were kept 2 decimals. (Lines 32-33)
M&M :
L83-87 : How many generations before experimentation were reared the aphids from the fields collection ?
Response: We clarify this in Materials and Methods, as such: “Aphis craccivora and M. japonica were reared on bean plantlets, and M. persicae were reared on pea plantlets until more than 30 generations in the greenhouse at 25 ± 1°C, 50 ± 5% RH and 16:8 (L: D) h”. (Lines 88-89)
L90-94 : Same question for the hoverfly species ? How many individuals did you collect during these trials or in other word how many individuals constituted your starting population ?
Response: Our starting population were 25 adults of E. corollae (♀:♂=15:10). (Line 92)
L98 : were raised individually in Petri dishes? please specify it.
Response: Done. Newly hatched larvae of E. corollae were raised with A. craccivora, M. persicae or M. japonica individually in a petri dishes (3.5 cm diameter × 1 cm height), with 100 aphids added daily until pupation.(Lines 100-102)
L100 : how many aphids ?
Response: Our pre-experiment found thatthe 3rd instar larvae of E. corollae can feed on up to 90 A. craccivora, 80 M. persicae or 40 M. japonica per day. Thus, newly hatched larvae of E. corollae were raised individually in a petri dishes with 100 aphids added daily until pupation. (Line 102)
L138-139 : Prior to use the ANOVA, do you test the normality of each variables ? If not, you might violate assumption conditions of the ANOVA and in this case needs to pass to a non parametric test (such as Kruskal-Wallis).
Response: Done. We have now changed the method of data analysis. Data analysis has now been adapted, as such: “Differences in the flight ability variables and pupal mass were analyzed using non-parametric Kruskal-Wallis test followed by a Bonferroni-adjusted significance test for pairwise comparison at the 0.05 significance level. Differences in larval survival, pupation rate, emergence rate, and hatching rate of egg were determined using a one-way analysis of variance (ANOVA) followed by Tukey’s honestly significant dif-ference (HSD), with proportional data first arcsine square-root-transformed to meet the assumptions of normality and heteroscedasticity. Differences in the duration of developmental stages between genders were determined using Student’s t-test, and differences in survival curves of E. corollae after feeding on different aphids were ana-lyzed using a log rank test. Paired bootstrap test with 100,000 replications was used for the precise estimation of the mean and standard error among the demographic pa-rameters and developmental duration of 3 diet treatments. All tests were performed in the program SPSS version 25 (IBM, Armonk, NY, USA), except life table variables were calculated and differentially analyzed in TWO-SEX-MSChart (Chi 2019) and plotted in OriginPro 2021 (OriginLab Corporation, Northampton, MA, USA).” (Lines 140-154)
L139 : Justify why to use the "Tukey’s honestly significant difference (HSD)" test ?
Response:
L140 : On which variables do you apply proportional data, is it on mx ?
Response 8: Thanks for your question, We apply proportional data on these variables (Table 2: larval survival rate (%), pupation rate (%), emergence rate (%), hatching rate of egg (%)), not including the “mx”. “mx” is the average population fecundity from egg to x days old, is not proportional data.
Results :
L153 : "different aphid treatments"
Response 9: Thank you. we have corrected. (Line 162)
L158 : "not"
Response 9: Thank you. we have corrected. (Line 166)
L160 : "pupae" ?
Response 9: Thank you for your careful review of my manuscript. We have changed "pupa" to "pupae". (Line 171)
Table 1 : it seems confusing with the different letters after mean± SE, could you please better explain than you did in the end of this table ?
Response 10: Done. This has now been adapted, as such: “Values are mean± SE. Different lowercase letters on the same row mean significant differences (paired bootstrap test, P<0.05). Means for male and female at the same developmental stage in same column followed by different capital letters differed significantly (Student’s t-test, P<0.05)”. (Lines 199-202)
Table 2 : I did not understand to which groups or individuals the number n = 3 is referring to ? Could you specify this ?
Response 11: For each diet, our sample consisted of a total of 90 eggs or newly hatched larvae (each diet treatment consisted of 30 eggs or newly hatched larvae and was replicated three times).
Figure 1 : Can we have SEM on each point of all the survival curves ?
Figure 2 : Can we have SEM on each point of all the survival curves ?
Response 12: Experimental data (90 eggs or larvae for each diet) were analyzed in the life table analysis software TWOSEX-MSChart (Chi 2019), the values without SEM of parameters (such as Sxj, lx, mx, lxmx , etc.) were obtained. And then we used these values to make graphs (Figure 1 and 2) in OriginPro 2021 (OriginLab Corporation, Northampton, MA, USA)..
Table 3-4-5 : To be coherent with the first tables, it would be interesting to invert lines and columns of these three tables.
Response 13: Thanks for your advice. Based on your comments, I have inverted lines and columns of these three tables. (Table 3-4-5)
Discussion :
L312-315 : I completely agree with this statement. In the conclusion part and the abstract, this could be put forward more.
Response 14: We have also highlighted this in the conclusions and the astracts (Lines 36-38 and Lines 352-354)
L325-329 : This part could move in the intro part, it does not add some improvements at this part of the discussion, it could also be deleted from the discussion.
Response 15: Done. As suggested, we have now deleted this part.
